# *Position:* Algebra Unveils Deep Learning
# An Invitation to Neuroalgebraic Geometry

**Giovanni Luca Marchetti** [* 1]   **Vahid Shahverdi** [* 1]   **Stefano Mereta** [* 1]   **Matthew Trager** [* 2]   **Kathlén Kohn** [* 1]

## Abstract

In this position paper, we promote the study of function spaces parameterized by machine learning models through the lens of algebraic geometry. To this end, we focus on algebraic models, such as neural networks with polynomial activations, whose associated function spaces are semi-algebraic varieties. We outline a dictionary between algebro-geometric invariants of these varieties, such as dimension, degree, and singularities, and fundamental aspects of machine learning, such as sample complexity, expressivity, training dynamics, and implicit bias. Along the way, we review the literature and discuss ideas beyond the algebraic domain. This work lays the foundations of a research direction bridging algebraic geometry and deep learning, that we refer to as neuroalgebraic geometry.

## 1. Introduction

Parametric machine learning models, such as neural networks, define a space of functions as their parameters vary. These spaces are often referred to as *neuromanifolds* (Kohn, 2024; Calin, 2020).

The geometry of neuromanifolds is intimately related to a number of questions at the heart of machine learning. Some geometric properties of neuromanifolds, such as their dimension, control statistical and computational aspects of the corresponding model, including sample complexity and expressivity. Moreover, neural networks learn through a process of gradient flow of their objective function. This optimization can be interpreted as minimizing a functional distance over the neuromanifold, effectively attracting the

---
[*]Equal contribution [1]Department of Mathematics, KTH Royal Institute of Technology, Stockholm, Sweden [2]AWS AI Labs, New York, USA (Work done outside Amazon). Correspondence to: Giovanni Luca Marchetti <glma@kth.se>.

*Proceedings of the $42^{nd}$ International Conference on Machine Learning*, Vancouver, Canada. PMLR 267, 2025. Copyright 2025 by the author(s).

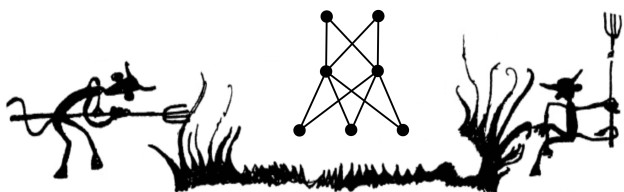

*Figure 1.* A neural variation of a celebrated doodle from the algebraic geometry literature (Grothendieck, 1968).

model towards an estimate of the ground-truth function. Consequently, geometric problems over neuromanifolds, such as nearest point problems, govern the training dynamics and provide insights into how neural networks learn.

Therefore, understanding the geometry of neuromanifolds offers a twofold potential. First, it serves as a powerful theoretical framework for analyzing and explaining empirical phenomena in machine learning. Second, it can lead to the design of novel machine learning architectures. By establishing precise relationships between architectural parameters and the resulting neuromanifold geometry, we can systematically design models that exhibit desired theoretical and practical properties.

### 1.1. The Power of Algebraic Geometry

A rich class of machine learning models are *(semi)-algebraic*, meaning that the corresponding function is (piece-wise) polynomial in both the input and the parameters. Examples include deep neural networks with polynomial or Rectified Linear Unit (ReLU) activation function.

For (semi-) algebraic models, the neuromanifold can be defined by a finite set of polynomial equalities and inequalities, i.e., it is a semi-algebraic *variety*. The study of these spaces is central to *algebraic geometry*, a field with a rich mathematical history that offers a wealth of ideas, tools, and invariants. Many of these, as we will argue, can be interpreted from the perspective of machine learning, shedding light on important questions in the field. For example, a fundamental invariant of algebraic varieties is their degree, which is characteristic of algebraic geometry since it

| Machine Learning | Algebraic Geometry |
|---|---|
| sample complexity and expressivity | dimension, degree, and covering number |
| subnetworks and implicit bias | singularities |
| identifiability and invariance | fibers of the parameterization |
| optimization and gradient descent | critical point theory, discriminants, and dynamical invariants |

*Table 1.* A dictionary between machine learning and algebraic geometry.

is undefined for, say, general differentiable manifolds. The degree plays a central role in learning aspects of the corresponding model (see Section 4.1). Another crucial advantage of algebraic geometry is that, unlike differential geometry, it naturally encompasses singular spaces. This is convenient in the context of deep learning; despite the 'manifold' terminology, neuromanifolds of neural networks are far from being smooth spaces, since they often exhibit singularities that affect the behavior of the corresponding model. Moreover, as previously mentioned, distances in the ambient space of neuromanifolds play a central role in the learning process of machine learning models. This suggests that the metric structure should be taken into account when analyzing neuromanifolds. Distance-based problems over algebraic varieties are the focus of the field of *metric algebraic geometry* (Breiding et al., 2024), from which we will borrow several tools throughout this work.

While focusing on algebraic models might seem restrictive, we note that arbitrary (continuous) functions can be approximated by algebraic ones with arbitrary precision. Thus, algebraic models are not only general, but can approximate arbitrary neuromanifolds. This allows for the extension of results and techniques discussed in this work to non-algebraic models, at least approximately. We will expand on this in Section 5.1.

### 1.2. Neuroalgebraic Geometry

Based on these reflections, we argue that **algebraic machine learning models can be studied using the powerful toolbox of algebraic geometry, leading to new insights in deep learning theory**. We propose the term *neuroalgebraic geometry* to refer to this emerging field of research. Neuroalgebraic geometry is closely related to algebraic statistics (Pistone et al., 2000) – the study of statistical models defined by polynomial equations – and can be interpreted as its counterpart in the context of machine learning.

In the following sections, we outline aspects of machine learning that can be reformulated in terms of (metric) alge-

braic geometry, highlighting natural intersections between the two fields. Along the way, we review and unify the previous literature in this direction. This work is intended as an invitation to the field of algebraic geometry for researchers from machine learning, and vice versa. From a broader perspective, we hope that it will inspire interdisciplinary research, positioning algebraic geometry among the mathematical disciplines that collaboratively contribute to unraveling the fundamental principles of deep learning.

## 2. Alternative Views

A potential counterargument to our position is that the relevance of algebraic models might be limited, both theoretically and in practice. Indeed, polynomials are rarely used as activation functions in real-life neural networks, and they are excluded by standard formulations of the universal approximation theorem for deep learning (Pinkus, 1999). However, we believe that algebraic models provide a setting where many intuitions that hold for networks in general can be formulated rigorously and, potentially, proved. Moreover, algebraic models are widely used as building blocks in actual architectures, as argued in Section 3. Regarding approximation results, we note that algebraic models can approximate arbitrary network architectures, and indeed universal approximation theorems can be extended to include polynomials by considering networks with varying depth (Yu et al., 2021).

From a broader perspective, several alternative approaches have been explored to analyze deep learning models, and in particular the function spaces they parameterize. Each of these approaches advocates for a specific mathematical formalism, which naturally brings original ideas to the field of machine learning, but also comes with restrictions on the models it considers. We believe that algebraic geometry provides unique tools to analyze aspects that are overlooked by other formalisms, as we will discuss in the next section.

## 2.1. Related Approaches

*Information geometry* (Amari, 2016) is a well-established field which studies statistical models and neuromanifolds from the perspective of Riemannian geometry. While similar in spirit to neuroalgebraic geometry — both study the geometry of neuromanifolds — the two approaches rely on different mathematical machinery. The Riemannian approach of information geometry is not constrained to algebraic models, but it fails to capture some fundamental aspects of neural models. As mentioned in Section 1, algebraic geometry provides richer invariants than differential and Riemannian geometry – such as the degree – and it is more suited for analyzing singular neuromanifolds. The importance of singularities for these models has led to the development of *singular learning theory* (Watanabe, 2009). The latter stems from information geometry, and naturally employs ideas from algebraic geometry. Neuroalgebraic geometry extends this direction, aiming to explore the full potential of a bridge between algebraic geometry and machine learning.

Another popular approach to the theoretical analysis of machine learning models leverages kernel methods – in particular, the *neural tangent kernel* (Jacot et al., 2018). The latter advocates for the study of neural networks at the infinite-width limit, where the model converges to kernel regressors. Questions of expressivity and statistical behavior can then be phrased in the language of functional analysis and Hilbert spaces, deriving insights from the infinite-dimensional linearized limit. In contrast, neuroalgebraic geometry considers finite-dimensional and nonlinear algebraic varieties that constitute neuromanifolds of algebraic models, which can approximate arbitrary neuromanifolds (see Section 5.1). Both approaches are therefore in a sense complementary to each other.

Further examples of approaches bridging pure mathematics – in particular, algebra and geometry – with machine learning include:

- geometric deep learning (Bronstein et al., 2021), concerned with symmetry properties of neural networks such as invariance and equivariance,

- topological data analysis (Edelsbrunner & Harer, 2010), aiming at extracting topological features from data,

- topological deep learning (Hajij et al., 2022), concerned with learning from data over non-Euclidean spaces, such as graphs and simplicial complexes,

- categorical deep learning (Gavranović et al., 2024), leveraging on category theory for a compositional understanding of neural architectures.

## 3. Preliminaries

In this section, we review some basic concepts from machine learning and introduce the main objects of study of neuroalgebraic geometry, i.e., neuromanifolds.

A *parametric machine learning model* is a mapping $\mathcal{W} \times \mathcal{X} \to \mathcal{Y}$ that associates parameters and inputs $(w, x)$ to an output $y$, denoted as $y = f_w(x)$. Given a dataset consisting of a finite collection of input-output pairs $\mathcal{D} \subset \mathcal{X} \times \mathcal{Y}$, the *empirical risk minimization (ERM)* is an optimization problem consisting in minimizing over $\mathcal{W}$ the following objective:

$$L_{\mathcal{D}}(w) := \frac{1}{|\mathcal{D}|} \sum_{(x,y) \in \mathcal{D}} \ell(y, f_w(x)), \qquad (1)$$

where $\ell : \mathcal{Y} \times \mathcal{Y} \to \mathbb{R}$ is a *loss function*. When $\mathcal{Y}$ is a Euclidean space or a probability simplex, respectively, common choices for $\ell$ are the *quadratic loss* $\ell(y, \hat{y}) = \|y - \hat{y}\|^2$ or the *cross-entropy loss* $\ell(y, \hat{y}) = -\sum_j y_j \log \hat{y}_j$. The goal of ERM is to find parameters that generalize to unseen examples. This is typically formulated by assuming that datapoints are drawn from some distribution, and that the empirical objective $L_{\mathcal{D}}$ approximates $\mathbb{E}_{x,y}[\ell(y, f_w(x))]$, referred to as *generalization error*. Often the data distribution is further assumed to take the form $x \sim \pi(x)$, $y = f^*(x)$, where $\pi$ is a distribution of inputs and $f^*$ is a deterministic function mapping inputs to outputs, interpreted as the ground-truth function underlying the given task.

In deep learning, sophisticated parametric models are constructed by composing simpler ones, referred to as modules or layers. The connectivity of such layers determines an architecture class – a family of models with similar design, e.g., a *Multi-Layer Perceptron (MLP)* is a sequential composition

$$f_w = W_L \circ \sigma \circ W_{L-1} \circ \sigma \cdots \circ W_1, \qquad (2)$$

where $w = (W_1, \ldots, W_L)$, $W_i$ is a linear or affine layer and $\sigma \colon \mathbb{R} \to \mathbb{R}$ is an *activation function* applied coordinate-wise. Other classes of modules include structured linear layers (e.g., those with sparse or convolutional weight matrices), normalization layers, and sequence-modeling modules such as (self-) attention mechanisms.

### 3.1. Neuromanifolds

**Definition 3.1.** *The neuromanifold of a parametric machine learning model $f$ is:*

$$\mathcal{M} := \{f_w \colon \mathcal{X} \to \mathcal{Y} \mid w \in \mathcal{W}\}. \qquad (3)$$

In other words, the neuromanifold is the image of the parametrization map, denoted by $\varphi \colon \mathcal{W} \to \mathcal{M}, w \mapsto f_w$.

Describing neuromanifolds is challenging, as even simple architecture classes parametrize complex families of functions. Indeed, the parametrization map is typically non-linear and not one-to-one, resulting in different parameters that identify the same function. Therefore, the neuromanifold carries intrinsic geometric structure that differs drastically from the one of the parameter space $\mathcal{W}$. As we shall see, the geometry of $\mathcal{M}$ is related to several fundamental aspects in machine learning.

In this work, we argue that the study of neuromanifolds is particularly appealing for *algebraic* models. Assuming that $\mathcal{W}$, $\mathcal{X}$ and $\mathcal{Y}$ are Euclidean spaces, we say that a parametric model $f$ is algebraic if it is a polynomial in both the parameters and the input. In this case, the neuromanifold contains vector-valued polynomials $f_w$ of bounded degree. In particular, $\mathcal{M}$ is contained in a finite-dimensional linear sub-space $\mathcal{V}$ of the function space. We refer to $\mathcal{V}$ as the *ambient space* of the neuromanifold. By the Tarski-Seidenberg theorem (Bierstone & Milman, 1988), the neuromanifold is a *semi-algebraic variety*, i.e., it can be characterized by polynomial equalities and inequalities in $\mathcal{V}$. The algebraic nature of these spaces allows for theoretical analysis using tools from algebraic geometry. Neuromanifolds of algebraic models are, therefore, the core focus of neuroalgebraic geometry.

The main class of examples of algebraic models is provided by neural networks with polynomial activation functions. Indeed, if $\sigma$ is a polynomial, then the expression in Equation 2 is polynomial in both $x$ and $w$. The study of MLPs with monomial activation function $\sigma(z) = z^k$ has been initiated by Kileel et al. (2019). For shallow networks – i.e., with $L = 2$ layers – the corresponding neuromanifold coincides with the space of symmetric tensors of bounded (Waring) rank (Arjevani et al., 2025). For linear networks – i.e., with no activation function or, equivalently, with $k = 1$ – the neuromanifold is, similarly, a space of matrices with bounded rank (Trager et al., 2020). This space is referred to as *determinantal variety* and it is well-understood – see Section A for a detailed discussion. For Convolutional Neural Networks (CNNs), the geometry of the neuromanifold differs drastically. In this case, up to rescaling the parameters, $\mathcal{M}$ is linearly birational – i.e., related by a linear map that is an isomorphism almost everywhere – to a more complex space known as the Segre-Veronese variety (Kohn et al., 2022; 2024; Shahverdi, 2024; Shahverdi et al., 2025a). A further example of an algebraic model is provided by linear (or 'lightning') attention mechanisms. These are non-normalized versions of the layers of a Transformer (Vaswani, 2017), which are popular nowadays across several applications. Linear attention mechanisms are cubical layers, and their neuromanifolds behave similarly to CNNs (Henry et al., 2025).

We remark that neuroalgebraic geometry can encompass, more generally, models that are piece-wise polynomial, fractions of such (Boullé et al., 2020), or even include roots. Such models also have semi-algebraic neuromanifolds. This includes neural networks with piece-wise linear activation functions, such as ReLU $\sigma(x) = \max\{0, x\}$. These models are shortly discussed in Section 5.2.

### 3.2. Learning on the Neuromanifold

From a geometric perspective, fitting a parametric model to data can be seen as a constrained optimization problem over the neuromanifold. However, as noted above, optimization in practice takes place in parameter space, leading to two related problems in $\mathcal{W}$ and $\mathcal{M}$ respectively. By denoting $L_\mathcal{D} = \mathcal{L}_\mathcal{D} \circ \varphi$, where $\mathcal{L}_\mathcal{D}$ is the extension of Equation 1 to the ambient space $\mathcal{V}$, we can write the following two equivalent optimization problems:

| **parameter space** | **function space** |
|:---:|:---:|
| $\min_{w \in \mathcal{W}} L_\mathcal{D}(w)$ | $\min_{f \in \mathcal{M}} \mathcal{L}_\mathcal{D}(f)$ |

The *loss landscape* is the graph of the loss $L_\mathcal{D}$ in parameter space. For algebraic loss functions $\mathcal{L}_\mathcal{D}$ (e.g., quadratic loss or Wasserstein distance for discrete distributions), the loss landscape is also a semi-algebraic variety, similar to the neuromanifold. For the cross-entropy loss or other loss functions with algebraic derivatives, at least the most important points of the loss landscape – namely, the critical points – depend algebraically on the data $\mathcal{D}$. Thus, in these settings, training is an algebraic optimization problem.

While the optimization in parameter space benefits from a Euclidean domain, enabling the use of algorithms such as gradient descent, the optimization problem in function space $\mathcal{V}$ is often more tractable from a mathematical perspective. In the algebraic setting, this problem is a polynomial program constrained to the neuromanifold. These programs have a well-developed theory, including, e.g., hierarchies of simplified relaxations (Lasserre, 2001). Moreover, optimization in function space can be often interpreted geometrically. For the quadratic loss, ERM consists in minimizing a potentially-degenerate quadratic form over $\mathcal{M}$, while the corresponding generalization error coincides with the squared $L^2$ distance between $f \in \mathcal{M}$ and the ground-truth function $f^*$. In both cases, learning amounts to a nearest-point problem over the neuromanifold w.r.t. the (squared) distance associated to the quadratic form. As anticipated in Section 1, this type of problems lies at the heart of metric algebraic geometry, whose tools, as we shall see, are central to neuroalgebraic geometry.

# 4. Deep Learning from an Algebro-Geometric Perspective

In this section, we discuss tools, ideas, and invariants from algebraic geometry relevant to deep learning. The topics covered are summarized in the dictionary in Table 1. Along the way, we review the existing literature. In Appendix Section A, we discuss a toy example of a network where everything covered in this section can be explicitly described and interpreted from a machine learning perspective.

## 4.1. Dimension, Degree, and Covering Number

The notion of **dimension** is, perhaps, the most natural numerical invariant of a space. The dimension of an algebraic variety can be defined as the linear dimension of its generic local linearization, i.e., the maximum number of independent vectors in the tangent space at a generic point. For neuromanifolds, the dimension reflects the intrinsic degrees of freedom of the model and is a simple measure of expressivity that is more precise than the parameter count. In an algebraic setting, the dimension of a neuromanifold can be computed exactly using symbolic methods.

On the other hand, the **degree** of a variety is an algebraic measure of how 'curved' it is in its ambient space. More precisely, the degree $d$ is the number of complex intersection points between the variety and a generic affine subspace of complementary dimension (appealing to complex intersection points is necessary for such number to be well-defined). When the variety is defined by a single polynomial equation, its degree coincides with the degree of that polynomial, while for general varieties the situation is more involved, yet the degree is in principle computable using symbolic methods.

Together, dimension and degree provide bounds on metric invariants of a variety. Specifically, for a compact variety $\mathcal{M}$ equipped with a metric, they bound the **covering number** $\mathcal{N}_\varepsilon(\mathcal{M})$, defined as the minimum number of metric balls of radius $\varepsilon$ required to cover $\mathcal{M}$ (Figure 2). Indeed, the covering number satisfies (Kileel et al., 2019):

$$\log \mathcal{N}_\varepsilon(\mathcal{M}) = \mathcal{O}\left(m \log \frac{d}{\varepsilon} + C\right), \qquad (4)$$

where $m$ and $d$ are the dimension and the degree of $\mathcal{M}$, respectively, and $C$ is a constant depending on the dimension of the ambient space $\mathcal{V}$ but independent of $m, d$, and $\varepsilon$. This type of bounds for covering numbers originated from the theory of Vitushkin variations (Yomdin & Comte, 2004) – higher-order analogues of the degree, capturing more meticolously the 'twistedness' of the variety.

Now, covering numbers can be interpreted as a measure of capacity, and are related to fundamental aspects in machine learning. First, they provide an elementary bound on the

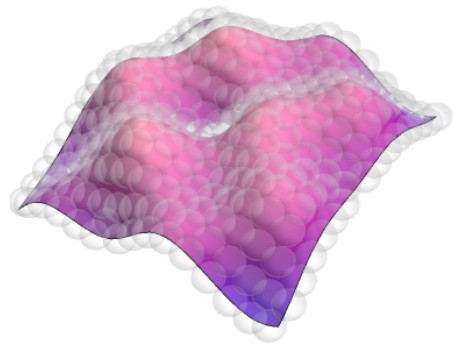

*Figure 2.* A manifold covered by balls.

volume of the **tubular neighborhood** $\mathcal{M}_\varepsilon$ consisting of points in the ambient space of $\mathcal{M}$ at a distance less than $\varepsilon$ from a point in $\mathcal{M}$:

$$\mathrm{Vol}(\mathcal{M}_\varepsilon) \leq \mathcal{N}_\varepsilon(\mathcal{M})\, \omega_{2\varepsilon}, \qquad (5)$$

where $\omega_{2\varepsilon}$ is the volume of a ball of radius $2\varepsilon$. When $\mathcal{M}$ is a neuromanifold, the volume of $\mathcal{M}_\varepsilon$ measures the set of functions that can be approximated by a model within a (quadratic) error of $\varepsilon$, which can be seen as a form of **approximate expressivity**. The combination of Equation 5 with Equation 4 is a modern formulation of bounds for tubular volumes (Basu & Lerario, 2023), but the history of the result goes back to Weyl's Tube Formula in Riemannian geometry (Weyl, 1939). Second, covering numbers of neuromanifolds play a central role from a statistical perspective due to their relation to the **sample complexity** of learnability. Roughly speaking, according to a fundamental result in statistical learning theory (Cucker & Smale, 2002; Pontil, 2003), the number of samples $|\mathcal{D}|$ required to infer the function that best approximates the distribution of data (with high probability, and within a given generalization loss margin $\varepsilon$) scales logarithmically in $\mathcal{N}_\varepsilon(\mathcal{M})$. By the discussion above, this bridges sample complexity to dimension and degree of the neuromanifold – we provide a geometric intuition on this relation in Section B. In conclusion, covering numbers control both the expressivity and sample complexity of the corresponding model, establishing a fundamental trade-off between them.

---

### Takeaway

*The dimension and degree are the most fundamental invariants of an algebraic neuromanifold. They control metric quantities such as covering numbers, which in turn measure approximate expressivity and sample complexity of the model.*

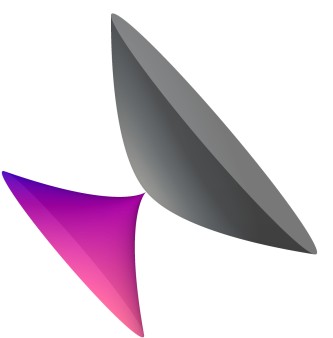

*Figure 3.* A singularity of a surface (pink) with a 3-dimensional Voronoi cell (gray).

## 4.2. Singularities

As mentioned in Section 1.1, algebraic varieties differ from differentiable manifolds in that they may exhibit **singularities**. A point on a variety is called singular if its tangent space has a higher dimension than that of a generic point. Singularities manifest in various forms such as self-intersections or sharp cuspidal edges.

Singularities of the neuromanifold play an important role in the training process of machine learning models, as highlighted in singular learning theory (Watanabe, 2009). Indeed, the learning dynamics can be attracted (locally) by singularities. This introduces an **implicit bias** into the training process, favoring certain functions regardless of the data. In order to illustrate this, consider the minimization of the Euclidean distance from a point in the ambient space to a variety $\mathcal{M}$, a problem closely related to ERM with quadratic loss (see Section 3). The **Voronoi cell** of a point $f \in \mathcal{M}$ is the set of points in the ambient space $\mathcal{V}$ whose unique closest point on $\mathcal{M}$ is $f$. At a non-singular point, the Voronoi cell is contained in the normal space, i.e., the orthogonal complement of the tangent space. On the other hand, at a singular point, the Voronoi cell can be larger, and its dimension can exceed the co-dimension of $\mathcal{M}$ (see Figures 3 and 4). In this case, the singular point is more likely to solve the optimization problem. The concept of Voronoi cells and their relation to singularities extends beyond quadratic losses, e.g. to Wasserstein distances (Becedas et al., 2024) or cross-entropy (Alexandr & Heaton, 2021).

In addition to exhibiting large Voronoi cells, singular points also affect the training dynamics. They can act as attractors or slow down the optimization flow, a phenomenon that has been studied in information geometry (Amari et al., 2006).

Interestingly, singularities of neuromanifolds frequently correspond to **subnetworks** of the original architecture, i.e., functions that can be represented by a smaller network from the same model class. This type of property

has been observed for different architectures such as MLPs – with (Shahverdi et al., 2025b; Arjevani et al., 2025) and without (Trager et al., 2020) activation function – and for CNNs with polynomial activation functions (Shahverdi et al., 2025b). In these settings, singularities may be responsible for a form of 'automatic model selection', encouraging simpler models over more complex ones. This is also consistent with empirical observations in the literature, according to which the performance of a model can often be matched by sparse subnetworks obtained through pruning or distillation (Zhu et al., 2024) or with careful initialization (Frankle & Carbin, 2019).

---

### Takeaway

*Singularities of the neuromanifold can introduce implicit biases in the learning process. In deep learning, they often correspond to subnetworks, favoring the selection of simpler models.*

---

## 4.3. Parameterization and Fibers

So far, we have discussed the intrinsic geometry of the neuromanifold. Since the parameter space has trivial geometry, the complexity of the neuromanifold emerges from the parameterization map $\varphi \colon \mathcal{W} \to \mathcal{M}$. For instance, in parameter space, the singularities of the neuromanifold are smoothed out. Instead, they are captured as the critical points of the parameterization map (discussed in Section 4.4) or its (non-generic) **fibers**, which we focus on now.

The fibers of a map $\varphi \colon \mathcal{W} \to \mathcal{M}$ describe domain points that are collapsed to the same output. Formally, the fiber of $f \in \mathcal{M}$ is $\varphi^{-1}(f) = \{w \in \mathcal{W} \mid \varphi(w) = f_w = f\}$. For an algebraic map $\varphi$, almost all its fibers resemble each other (over $\mathbb{C}$), and are collectively referred to as the *generic fiber*. Computing the fibers of a network's parameterization formalizes the problem of **identifiability** in the context of machine learning, which is concerned with recovering the parameters of a model from the function they define.

Understanding the generic fiber of a network's parameterization $\varphi$ provides a practical way of computing the dimension of the neuromanifold, whose importance has been discussed in Section 4.1. According to the **fiber-dimension theorem** (Shafarevich & Hirsch, 1994, Theorem 1.25), the dimension of the image $\mathcal{M}$ coincides with the co-dimension, in the parameter space $\mathcal{W}$, of the generic fibers of the map $\varphi$.

Fibers can be related to the **invariance** properties of the model, which lie at the heart of geometric deep learning (Section 2.1). Many models carry a (linear) action by the general linear group $\mathrm{GL}(\mathbb{R}, \dim(\mathcal{X}))$ on their parameter

space $\mathcal{W}$ that satisfies the following adjunction property:

$$f_w(Tx) = f_{T \cdot w}(x) \qquad (6)$$

for all $T \in \mathrm{GL}(\mathbb{R}, \dim \mathcal{X})$ and $w \in \mathcal{W}$. For instance, for MLPs (Equation 2), this action modifies the first-layer parameters as $T \cdot W_1 = W_1 T$. Equation 6 implies that $f_w$ is invariant to a transformation $T$ of its input space if, and only if, $w$ and $T \cdot w$ belong to the same fiber. This principle has been exploited to describe invariant neural networks via group-theoretical harmonic analysis (Marchetti et al., 2024).

As an example, we now describe the fibers of MLPs. Other architectures have been considered, e.g. CNNs (Shahverdi et al., 2025b) and attention mechanisms (Henry et al., 2025). First, arbitrary MLPs exhibit combinatorial symmetries in their parameterization, by permuting the output of a layer and permuting back the input of the following one. Formally, given a permutation matrix $T$ for some layer $1 \le i < L$, the simultaneous transformations $W_i \mapsto TW_i$ and $W_{i+1} \mapsto W_{i+1}T^{-1}$ do not alter the function defined by the MLP. Moreover, for (positively) homogeneous activation functions $\sigma$ – e.g., $\sigma(x) = x^r$ or ReLU – a similar symmetry arises by rescaling the input and output of each neuron. Lastly, when $\sigma(x) = x$ is the identity, the same holds for arbitrary invertible matrices $T$. Intuitively, non-trivial activation functions 'break' symmetries, reducing the latter to neuron-wise transformations, i.e., permutations and possibly rescalings. It has been conjectured that such symmetries characterize the generic fibers of algebraic MLPs (Kileel et al., 2019; Kubjas et al., 2024), which has been recently proven for polynomial activations of high degree $r \gg 0$ (Finkel et al., 2024; Shahverdi et al., 2025b). In the non-algebraic case, the same result is well-known when $\sigma$ is the hyperbolic tangent (Fefferman et al., 1994), while for ReLU additional non-trivial symmetries arise (Grigsby et al., 2023).

---

### Takeaway

*Fibers of the parameterization control the dimension and symmetries of the neuromanifold. Together with critical points, they explain the singularities of the neuromanifold.*

---

### 4.4. Critical Points and Gradient Descent

In this section, we focus on the dynamics of learning and, in particular, on the equilibria of gradient descent, which are the **critical points** of the loss in parameter space:

$$L_{\mathcal{D}} \colon \mathcal{W} \xrightarrow{\varphi} \mathcal{M} \xrightarrow{\mathcal{L}_{\mathcal{D}}} \mathbb{R}. \qquad (7)$$

These aspects are intimately linked with the geometry of the neuromanifold (e.g., its singularities; see Section 4.2)

and its parameterization map. For instance, a simple observation is that an algebraic neuromanifold $\mathcal{M}$ may have three qualitative geometric behaviors: it can coincide with its ambient space $\mathcal{V}$, be a full-dimensional proper subset of $\mathcal{V}$, or lie within a lower-dimensional algebraic subset. In the first two scenarios, assuming a convex functional $\mathcal{L}_D$, all critical points of the loss that are not global optima are contained in the set $\mathrm{Crit}(\varphi)$ of critical points of the parametrization $\varphi$.

In general, a parameter $w \in \mathrm{Crit}(\varphi)$ can be a critical point of the loss $L_{\mathcal{D}}$ even though the corresponding function $f_w \in \mathcal{M}$ is *not* a critical point for the loss functional $\mathcal{L}_{\mathcal{D}}$ in function space (see Section A for an example). These parameters are sometimes called **spurious critical points** (Trager et al., 2020). For some network architectures, spurious critical points will be typically avoided by gradient descent, while for others they are found with positive probability (Kohn et al., 2022). For instance, in the case of monomial activation, MLPs can have spurious critical points as local minima with positive probability (Kileel et al., 2019), while for CNNs they only correspond to the zero function $0 \in \mathcal{M}$ (Shahverdi et al., 2025a). For attention networks, the characterization of spurious critical points is an open problem.

The non-spurious critical points of the loss can be studied directly on the neuromanifold $\mathcal{M}$. As we have already discussed the singularities of $\mathcal{M}$ in Section 4.2, we focus here on the critical points of $\mathcal{L}_{\mathcal{D}}$ on the smooth locus of $\mathcal{M}$. These are intimately related to the topology and geometry of $\mathcal{M}$ via **Morse theory** (Milnor, 1963). The latter establishes a connection between global topological invariants – such as Betti numbers – to critical points and their types. For instance, Morse theory can be applied to the squared Euclidean distance over algebraic neuromanifolds, i.e. critical points of $\|\cdot - f^*\|^2 \colon \mathcal{M} \to \mathbb{R}_{\ge 0}$, where $\mathcal{V}$ is the ambient vector space and $f^* \in \mathcal{V}$. This optimization problem is closely related to the training dynamics of the quadratic loss (Section 3). When this problem is complexified, the number of critical points for generic $f^*$ is finite and constant, defining an invariant of $\mathcal{M}$ deemed **Euclidean distance degree** (Draisma et al., 2016). This invariant is similar in spirit to the degree (Section 4.1), and intuitively quantifies the complexity of the distance minimization problem over $\mathcal{M}$. Over the real numbers, the Euclidean distance degree provides an upper bound for the number of critical points. More specifically, the number of real critical points and their type stay locally constant as $f^*$ varies in $\mathcal{V}$, and it changes only when crossing particular algebraic varieties in $\mathcal{V}$, deemed **data discriminants** (Breiding et al., 2024; Arjevani et al., 2025); see Figure 4. The concepts of Euclidean distance degree and data discriminants extend to general algebraic optimization problems beyond quadratic losses, such as Wasserstein distances (Çelik et al., 2020;

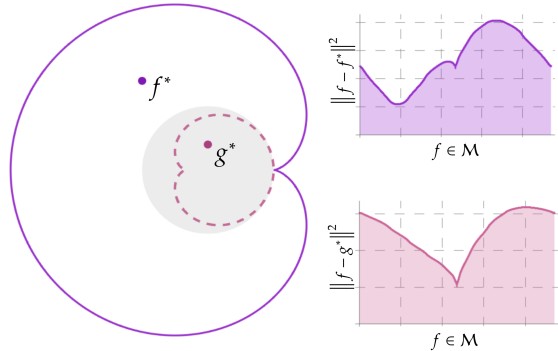

*Figure 4.* Illustration of a curve with the corresponding data discriminant (dashed) and Voronoi cell of the singularity (gray). For points inside (resp. outside) the discriminant, the Euclidean distance from the curve has 2 (resp. 4) real critical points.

Meroni et al., 2024) and cross-entropy loss (Catanese et al., 2006). From the perspective of machine learning, all these notions from (metric) algebraic geometry provide tools to study the behavior of critical points of the loss function and the corresponding optimization landscape.

While critical points describe the equilibria of the gradient flow, its dynamical behavior can be described by its preserved quantities, referred to as **dynamical invariants**. These are relations in the parameters $w \in \mathcal{W}$ that stay constant along the curve that gradient flow traces in $\mathcal{W}$. Dynamical invariants have been discussed for a variety of network architectures (Kohn et al., 2022; Williams et al., 2019; Du et al., 2018), and the algebraic ones have been recently described in detail (Marcotte et al., 2024). Knowing the invariants not only enables to analyze the training dynamics, but can be exploited to design well-behaved parameter initializations, since gradient flow will avoid all parameters that do not satisfy the same invariants as the initial values (Arora et al., 2018b). For instance, a balanced initialization can lead to stable learning dynamics.

---

### Takeaway

*The critical points of the loss are influenced by the geometry of the neuromanifold. Their number and type can change suddenly as data crosses discriminants. Moreover, algebraic invariants of gradient flow govern the training dynamics.*

---

## 5. Beyond the Algebraic

Throughout this work, we have focused on algebraic models. In this section, we discuss the possibility of applying similar ideas beyond the purely algebraic domain.

In our view, borrowing ideas from different fields of mathematics contributes to a holistic and multidisciplinary theory, which is essential for a fundamental understanding of deep learning. In fact, some mathematical ideas discussed in this work extend beyond the boundaries of algebraic geometry. For example, invariants such as dimension and covering number are defined for arbitrary metric spaces, singularities can be studied beyond algebraic varieties, the fiber-dimension theorem holds in great generality, and Morse theory is often formulated within the context of (smooth) manifolds. As a consequence, these general tools are applicable to non-algebraic models. Yet, they often result in a stronger incarnation when specialized to the algebraic context. For instance, the study of singularities is more tractable for algebraic models and, in addition to general Morse theory, algebraic geometry provides new concepts such as data discriminants and Euclidean distance degrees, enabling a more explicit understanding of loss landscapes.

In addition, we underline that ideas from algebraic geometry can be applied to non-algebraic models. Below, we discuss scenarios where such extensions of the methods of neuroalgebraic geometry are possible.

### 5.1. Polynomial Approximations

As anticipated in Section 1.1, general machine learning models can be approximated by algebraic ones. Specifically, by the Weierstrass Approximation Theorem (de la Cerda, 2023), polynomials are dense in the space of continuous functions over a compact space (with the uniform topology). This means that any continuous function can be approximated, at least locally, by polynomials, which can be exploited to approximate general neuromanifolds with algebraic ones.

More precisely, consider the example of an MLP with a continuous activation function $\sigma$, whose neuromanifold is denoted by $\mathcal{M}$. By restricting both the parameter space $\mathcal{W}$ and the input space $\mathcal{X}$ to compact subspaces, the output of every layer will be confined in a compact subspace as well. This means that $\sigma$ can be approximated (w.r.t. the uniform distance) over an appropriate closed interval of $\mathbb{R}$, obtaining an approximation (w.r.t. the Hausdorff distance) of $\mathcal{M}$ by the neuromanifold of an algebraic model.

Therefore, it is possible, in certain cases, to extend results for algebraic neuromanifolds – to which the techniques described in this paper apply – to general (continuous) ones via approximation arguments. As an example, this strategy has been applied to provide bounds on covering numbers of MLPs with ReLU activation function by exploiting the bounds for algebraic models discussed in Section 4.1 (Zhang & Kileel, 2023).

## 5.2. Tropical Geometry

For MLPs with ReLU activation function – and, more generally, with piece-wise linear ones – the associated neuromanifold can be studied directly via an alternative version of algebraic geometry, i.e. *tropical geometry* (Maclagan & Sturmfels, 2021). The latter is based on an algebra where the standard operations of addition and multiplication in $\mathbb{R}$ are replaced by maximum (or minimum) and addition, respectively. This gives rise to a geometric theory which is 'degenerate', in a certain (precise) sense. Polynomial functions in this theory are continuous convex piece-wise linear functions. The connection between ReLU networks and tropical geometry has been established by Arora et al. (2018a) and, more explicitly, by Zhang et al. (2018), where it is argued that any piece-wise linear function can be represented by such a network with enough layers.

Tropical geometry is closely related to combinatorics and, in particular, polyhedral geometry. This offers a plethora of tools from discrete mathematics to address geometric questions. In particular, the properties on the left-hand side of Table 1 can be studied by investigating the structure of combinatorial objects associated to neural networks, such as polytopes and fans (Montúfar et al., 2022; Brandenburg et al., 2024b). For example, recent works have explored these techniques to describe the fibers of the parameterization (Section 4.3) of ReLU networks (Brandenburg et al., 2024a), which relates to profound questions on decompositions of tropical rational functions (Tran & Wang, 2024).

---

Takeaway

*Algebraic methods can be applied beyond the polynomial domain, e.g. via approximation or by leveraging on alternative algebras.*

---

## 6. Conclusions and Future Directions

In this work, we have outlined the principles of neuroalgebraic geometry – an emerging field concerned with the geometric study of neuromanifolds of algebraic machine learning models. We have discussed several connections between algebraic geometry and machine learning, showcasing how problems from the latter can be rephrased in the language of the former and addressed mathematically. Our hope is that this invitation to neuroalgebraic geometry will foster interdisciplinary research between algebraic geometry and deep learning, unveiling mathematical principles underlying neural networks and their learning processes.

Throughout this work, we have discussed several open problems and general questions in the research program of neuroalgebraic geometry. We conclude by briefly outlining a few concrete directions for the immediate future. A natural next step is to extend the algebro-geometric analysis to a broader class of neural architectures. For instance, skip connections are algebraic components widely deployed in modern machine learning. Incorporating skip connections might result in a smoothing of neuromanifold singularities (Orhan & Pitkow, 2018), making their analysis particularly appealing from the perspective of neuroalgebraic geometry. Similarly, it would be interesting to understand the neuroalgebraic geometry of architectures such as Graph Neural Networks (GNNs) and State Space Models (SSMs), which are popular in several domains, and are closely related to attention-based and convolutional networks. These models incorporate structures in data such as symmetries and recursion, falling into the scope of geometric and topological deep learning (see Section 2.1). Even further, understanding how structures in data can be reflected by geometric features of the neuromanifold represents a more general and fundamental question.

From a broader perspective, a desirable outcome of neuroalgebraic geometry is to suggest the development of novel models, as mentioned in Section 1. We believe that the power of algebraic geometry and its tools lies not only in providing a descriptive framework, but also in offering prescriptive principles for the design of theoretically-grounded models. Potentially, this could pave the way to new generations of neural architectures whose behavior is controlled by algebro-geometric invariants.

## Acknowledgements

We thank Antonio Lerario for insightful discussions and Tomas Pajdla for helpful comments improving an earlier version of this manuscript. This work was partially supported by the Wallenberg AI, Autonomous Systems and Software Program (WASP) funded by the Knut and Alice Wallenberg Foundation.

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

# A. The Simplest Example

In this section, we discuss a toy example of a neural network with a simple, well-understood neuromanifold. In this case, several of the points discussed in Section 4 can be described explicitly and interpreted from a machine learning perspective.

We consider a linear MLP with two layers. Formally, given positive integers $N_0, N_1, N_2$, the model is defined as:

$$f_w(x) = W_1 W_0 \, x, \tag{8}$$

where $w = (W_0, W_1) \in \mathbb{R}^{N_1 \times N_0} \oplus \mathbb{R}^{N_2 \times N_1}$ is a pair of matrices. Since $f_w$ is linear, the ambient space of the neuromanifold is given by matrices $\mathcal{V} = \mathbb{R}^{N_2 \times N_0}$, where the neuromanifold $\mathcal{M}$ consists of the ones that factorize in two matrices as in Equation 8. When $N_1 \geq \min\{N_0, N_2\}$, the neuromanifold coincides with all linear maps, i.e., $\mathcal{M} = \mathcal{V}$. Instead, when the network exhibits a 'bottleneck' in its architecture, the neuromanifold consists of matrices of rank at most $N_1$. The latter is a well-understood space, deemed *determinantal variety* (Bruns & Vetter, 2006).

The determinantal variety can be described as the simultaneous vanishing locus of the determinants of all $(N_1+1) \times (N_1+1)$ matrix minors, which are polynomial equations in $\mathcal{V}$. It is therefore an algebraic variety. As mentioned in Section 4.3, the parameterization is invariant to the action by $T \in \mathrm{GL}(\mathbb{R}, N_1)$ defined as $W_0 \mapsto TW_0$, $W_1 \mapsto W_1 T^{-1}$. For generic $w$, this action provides an isomorphism between its fiber and $\mathrm{GL}(\mathbb{R}, N_1)$. As a consequence, the dimension of $\mathcal{M}$ is:

$$\dim(\mathcal{M}) = \dim(\mathcal{W}) - \dim(\mathrm{GL}(\mathbb{R}, N_1)) = N_1(N_0 + N_2 - N_1). \tag{9}$$

The degree of the determinantal variety is subtle to compute, and coincides with (Fulton, 2013):

$$\deg(\mathcal{M}) = \prod_{0 \leq i < \min\{N_0, N_2\} - N_1} \frac{\binom{\max\{N_0, N_2\}+i}{N_1}}{\binom{N_1+i}{N_1}}. \tag{10}$$

The singular points of the determinantal variety correspond to matrices with rank $< N_1$. From a machine learning perspective, these can be interpreted as functions defined by a network with a smaller architecture, where the width of the hidden layer has been reduced. This showcases the relation between singularities and subnetworks discussed in Section 4.2. Lastly, we consider the distance minimization problem over $\mathcal{M}$ with respect to the Euclidean distance over $\mathcal{V}$, i.e., the Frobenius distance between matrices. This is equivalent to the traditional problem of low-rank matrix approximation (Markovsky, 2012), which is ubiquitous across applications. The solution is well-known: by the Eckart-Young-Schmidt Theorem, the critical points of the distance function from $W \in \mathcal{V}$ over (the smooth locus of) $\mathcal{M}$ are obtained by projecting to $N_1$ singular values of $W$. The minima (resp. maxima) occur for the largest (resp. smallest) singular values. In particular, for generic $W$ there exist a unique minimum, and the Euclidean distance degree of $\mathcal{M}$ is $\binom{\min\{N_0, N_2\}}{N_1}$.

Finally, we compute the spurious critical points of a concrete linear two-layer MLP. We consider the example $N_0 = N_1 = N_2 = 2$, where the neuromanifold $\mathcal{M}$ is all of $\mathbb{R}^{2 \times 2}$, together with the loss functional $\mathcal{L}(f) = \|f - I\|^2$, i.e., the squared Frobenius distance from the identity matrix $I$. Since $\mathcal{M}$ is a vector space, $\mathcal{L}$ has a unique critical point over $\mathcal{M}$, given by $f = I$. However, the corresponding loss function $L$ in parameter space has several critical points, forming an algebraic variety. This variety has 3 irreducible components: the 4-dimensional fiber $\varphi^{-1}(I)$, the zero point $(0, 0)$, and a 4-dimensional component of spurious critical points (that is strictly contained in the locus where both $W_0$ and $W_1$ have rank 1). One concrete point in the latter component is given by $W_0 = \begin{bmatrix} 1/2 & 1/2 \\ 1 & 1 \end{bmatrix}$ and $W_1 = \begin{bmatrix} 1 & 0 \\ 1 & 0 \end{bmatrix}$.

# B. A Geometric Perspective on Sample Complexity

As discussed in Section 4.1, the dimension and the degree of the neuromanifold control the sample complexity of the corresponding model. In what follows, we provide a simple geometric intuition around this relation, in some specific scenarios. We consider the ERM problem where the dataset $\mathcal{D} \subset \mathcal{X} \times \mathcal{Y}$ is sampled from a ground-truth function $f^*$, and for simplicity assume that $\mathcal{Y} = \mathbb{R}$, i.e., the model is scalar-valued. In this case, each datapoint $(x, y)$ defines a hyperplane $H_{x,y}$ in the ambient space $\mathcal{V}$ consisting of functions $f$ passing through it, i.e., $f(x) = y$. The intersection

$$\mathcal{I} = \bigcap_{(x,y) \in \mathcal{D}} H_{x,y} \tag{11}$$

is the set of functions interpolating $\mathcal{D}$, and assuming the loss function $\ell$ is positive definite – such as the quadratic loss – it coincides with the vanishing locus in $\mathcal{V}$ of $\mathcal{L}_\mathcal{D}$. For generic $\mathcal{D}$, $\mathcal{I}$ is an affine subspace of $\mathcal{V}$ of co-dimension equal to the dataset size $n := |\mathcal{D}|$ – see Figure 5 for an illustration.

First, we consider the *realizable* scenario, meaning that $f^* \in \mathcal{M}$. The goal of the model is recovering $f^*$ from the data. Denote by $m$ the dimension of the neuromanifold $\mathcal{M}$. When $n < m$, the intersection $\mathcal{I} \cap \mathcal{M}$ is infinite, resulting in impossibility of determining $f^*$. If $n = m$, the intersection consists, for generic $\mathcal{D}$, of finitely many isolated points bounded by the degree $d$ of $\mathcal{M}$. Finally, when $n > m$, the intersection contains only $f^*$ for generic $\mathcal{D}$, resulting in unambiguous recovery. This shows that the sample complexity of learnability is (linear in) the dimension $m$.

Next, we consider the scenario where $f^* \in \mathrm{Span}(\mathcal{M}) \subseteq \mathcal{V}$, i.e., the ground-truth function is a linear combination of functions parametrized by the model. This can be interpreted as the realizable scenario for a 'mixture of experts' associated with the given model (Jacobs et al., 1991). Now, a simple result in algebraic geometry states that the dimension of $\mathrm{Span}(\mathcal{M})$ is bounded by $d + m$ (Eisenbud & Harris, 1987). By reasoning as above, when $n \geq m + d$, for generic $\mathcal{D}$, $\mathcal{I}$ intersects $\mathrm{Span}(\mathcal{M})$ only in $f^*$, meaning that the function(s) in $\mathcal{M}$ minimizing the generalization error can be recovered unambiguously. This shows that the sample complexity of learnability is $\mathcal{O}(m + d)$, implying that both dimension and degree play a role in this case.

In general, $f^*$ belongs to an infinite-dimensional function space a priori. As mentioned at the end of Section 4.1, in order to relate sample complexity to dimension and degree – and, more fundamentally, to the covering number – it is necessary to reason statistically over $\mathcal{D}$ and invoke concentration inequalities, which lies at the heart of statistical learning theory. To this end, the following classical result follows from Hoeffding's inequality (Cucker & Smale, 2002; Pontil, 2003):

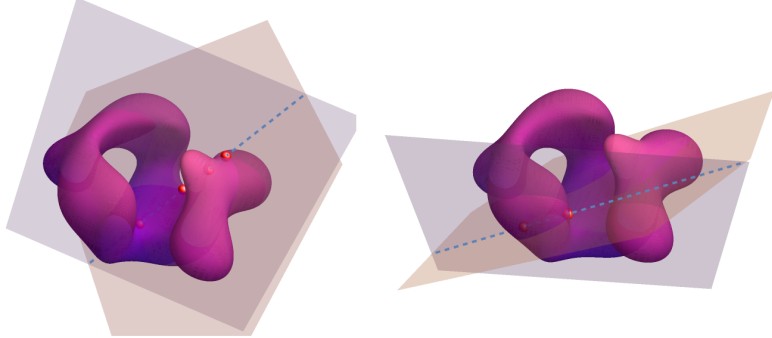

*Figure 5.* The intersection of a line (dashed) – identified by two (hyper-) planes – with a surface yields a finite number of solutions, bounded by the degree of the variety.

**Theorem B.1.** *Let $d$ be a metric on $\mathcal{V}$ and suppose that:*

- *there exists $C \in \mathbb{R}_{>0}$ such that for every probability distribution $\pi$ over $\mathcal{X} \times \mathcal{Y}$ the corresponding (generalization) error is $C$-Lipschitz, i.e., for every $f, g \in \mathcal{V}$:*

$$|\mathcal{L}_\pi(f) - \mathcal{L}_\pi(g)| \leq C\, d(f, g), \tag{12}$$

*where $\mathcal{L}_\pi(f) := \mathbb{E}_{(x,y) \sim \pi}[\ell(f(x), y)]$, and*

- *the loss is bounded on the neuromanifold $\mathcal{M} \subseteq \mathcal{V}$, i.e., there exists $D \in \mathbb{R}_{>0}$ such that $|\ell(f(x), y)| \leq D$ for all $f \in \mathcal{M}$ and $(x, y) \in \mathcal{X} \times \mathcal{Y}$.*

*Moreover, assume that $\mathcal{M}$ is compact. Then for every distribution $\pi$ over $\mathcal{X} \times \mathcal{Y}$ and every $n \in \mathbb{Z}_{>0}$, $\varepsilon \in \mathbb{R}_{>0}$:*

$$\mathbb{P}_{\mathcal{D} \sim \pi^n}\left(\sup_{f \in \mathcal{M}} |\mathcal{L}_\pi(f) - \mathcal{L}_\mathcal{D}(f)| \geq \varepsilon\right) \leq 2\, \mathcal{N}_{\frac{\varepsilon}{4C}}(\mathcal{M})\, e^{-\frac{n\varepsilon^2}{D^2}}, \tag{13}$$

*where $\mathcal{N}$ denotes the covering number w.r.t. $d$.*

As a consequence, for every $\varepsilon, \delta \in \mathbb{R}_{>0}$, if the dataset size satisfies

$$n = |\mathcal{D}| = \Omega\left(\frac{D^2}{\varepsilon^2} \log \frac{\mathcal{N}_{\frac{\varepsilon}{4C}}(\mathcal{M})}{\delta}\right), \tag{14}$$

then minimizing the empirical error $\mathcal{L}_{\mathcal{D}}$ (ERM) over $\mathcal{M}$ leads to the minimization of the generalization error $\mathcal{L}_{\pi}$ within a margin of $\varepsilon$ with probability at least $1 - \delta$. Put simply, the sample complexity of learning the function in $\mathcal{M}$ that minimizes the generalization error is logarithmic in the covering number of $\mathcal{M}$. Together with the connection between the (logarithmic) covering number with dimension and degree (Equation 4), this provides an upper bound on the sample complexity of algebraic models.

