# OpenReview forum: "Position: Algebra Unveils Deep Learning - An Invitation to Neuroalgebraic Geometry"
_ICML.cc/2025/Position_Paper_Track — ICML 2025 Position Paper Track spotlightposter_

### Official Review · Reviewer_gEu8 · 2025-02-13

**Significance:** 3
**Argument Clarity:** 3
**Rating:** 4
**Confidence:** 4

**Questions:**

- how could polynomial optimization be used to strengthen the current position?

**Discussion Potential:**

3

**Paper Summary:**

This paper advocates the need for considering neuroalgebraic geometry as a tool to understand fundamental aspects and empirical phenomena of machine learning, including sample complexity, expressivity and bias, and in addition as a tool to design novel models with the desired theoretical and practical features.

## update after rebuttal

The authors clearly emphasized in their rebuttal how polynomial optimization could be used to strengthen the current position. My score was already high (accept) before the rebuttal and I am now definitely convinced about keeping this high score.

**Position:**

Yes

**Position In Title:**

Yes

**Related Work:**

2

**Strengths And Weaknesses:**

Strengths:

- the advocated position is supported by the evidence that algebraic models allow one to approximate complex phenomena with a small number semi-algebraic functions
- it is likely to inspire discussion as neuroalgebraic geometry is a growing field
- the paper cites related works in an appropriate way

Weaknesses:

- the paragraph about polynomial approximation and tropical geometry is rather vague and could be improved
- the paper does not mention polynomial optimization and the related hierarchies of convex programs, a powerful tool for decision algorithms related to semi-algebraic functions.

**Support:**

3

---

> ### Author Rebuttal · Authors · 2025-03-31
>
> We thank the reviewer for the feedback and the appreciation.
>
> We wish to comment on the question raised regarding polynomial optimization. Training a machine learning model amounts to minimizing a loss over a semi-algebraic variety (i.e., the neuromanifold), which can be rephrased as a constrained optimization problem, with polynomial constraints. Therefore, as mentioned in the review, polynomial optimization is related to neuroalgebraic geometry – this is briefly mentioned in Section 2.2.
>
> However, solving the polynomial optimization problem directly is often infeasible for algebraic machine learning models. First, the constraints are typically polynomials of large degree in a high-dimensional space. Second, even computing the constraints is a challenging task. Since they coincide with the polynomials defining the neuromanifold, finding them explicitly requires computing the equations and inequalities guaranteed by the Tarski-Seidenberg theorem (Section 2.1). While algorithms to this end exist and are being developed [1], they are infeasible to scale to the dimensions and degrees involved in deep learning.
>
> Yet, we believe that the ideas from polynomial optimization can play an important role in the theoretical analysis of the training objective of algebraic machine learning models. For example, powerful tools such as the Lasserre’s sum-of-squares hierarchy [2] might provide insights on the solutions of the optimization problems arising in machine learning, since they allow to approximately rephrase these problems as simpler programs.
>
> We find that the connection between neuroalgebraic geometry and the field of polynomial optimization is an interesting research direction that strengthens the motivation for considering and studying algebraic models. As such, we will expand on it in Section 1.4 in the updated version of the manuscript.
>
>
> [1] Basu, Algorithms in real algebraic geometry: a survey, 2014.
>
> [2] Lasserre, Global optimization with polynomials and the problem of moments, 2001.

---

> > ### Comment · Reviewer_gEu8 · 2025-04-02
> >
> > I would like to thank the authors for their detailed reply. I will maintain my score.

---

### Official Review · Reviewer_Ynie · 2025-03-09

**Significance:** 4
**Argument Clarity:** 4
**Rating:** 4
**Confidence:** 4

**Questions:**

Could you comment on which results are restricted to semi-algebraic models and which ones are not?

**Discussion Potential:**

4

**Paper Summary:**

The paper describes different techniques and results from algebraic geometry useful for analyzing certain deep learning model classes.
For example, it claims that sample complexity and expressivity can be analyzed in terms of the dimensions and covering numbers of the neuromanifolds, implicit biases can be seen in terms of the geometry and singularities of the neuromanifold, among other results.
Table 1 connects machine learning questions with techniques from algebraic geometry.

**Position:**

Yes

**Position In Title:**

Yes

**Related Work:**

3

**Strengths And Weaknesses:**

This paper makes a great case for algebraic geometry as a technique to study deep learning models. It is well-written, and it refers to relevant papers.

The paper would be improved if it explained which ones of these techniques are restricted to semi-algebraic models and which ones would extend to smooth (or piece-wise smooth, or real analytic) functions (which include important functions for deep learning, such as the softmax). And when the results are only restricted to polynomials, it would be interesting to mention what the obstruction is to extend it to more general classes of functions. Many of these techniques are not restricted to polynomials (e.g. Morse theory, implicit bias results, etc) so the restriction of the whole paper to semi-algebraic machine learning models seems a bit artificial.

**Support:**

3

---

> ### Author Rebuttal · Authors · 2025-03-31
>
> We thank the reviewer for the comments and the appreciation. Below, we wish to answer to the question raised regarding the restriction to semi-algebraic models.
>
> We agree that some of the tools and results discussed in our work extend beyond the (semi-) algebraic realm. Among the points discussed in Section 3, we highlight the following ones, which hold in a greater generality:
>
> - Notions of **dimension** (Section 3.1) can be defined for general spaces, such as the Lebesgue dimension for topological spaces, or the Minkowski and Hausdorff dimensions for metric spaces. All these are related to covering numbers, and coincide with the usual dimension for differentiable manifolds and (semi-) algebraic varieties. Therefore, it is, in principle, possible to consider the dimension of neuromanifolds of arbitrary machine learning models, and to connect them to sample complexity as explained in Section 3.1 and in the appendix (Section B).
>
> - **Fibers** of the parameterization (Section 3.3), together with their connection to identifiability questions, are defined for arbitrary models, and have been considered in the literature. A seminal work in that direction is [1], which characterized the fibers of an MLP with tanh activation function. This work has been extended to more general sigmoidal activations [2].
>
> - As mentioned in the review, **Morse Theory** (Section 3.4) applies to general dynamical systems over smooth manifolds, and is traditionally considered within the general realm of (algebraic) topology.
>
> - **Implicit bias** to subnetworks (Section 3.2) makes sense for networks with arbitrary activation functions, and is in fact typically observed for non-algebraic models [3], which are more common in practice.
>
> On the other hand, some tools discussed in our work are specific and characteristic of algebraic geometry, as mentioned in the introduction (Section 1.1).  For example, the **degree** (Section 3.1) is an invariant defined only for algebraic varieties due to their polynomial nature, and is central in questions related to expressivity, as we discuss. Another example is provided by **singularities**, which are central in (singular) learning theory. In the algebraic context, it is much more feasible to explicitly study the type of singularities and their effect on optimization. Although implicit bias makes sense for general models, for algebraic ones it can be explained rigorously via singularities.
>
> We wish to add that even though some notions generalize beyond the algebraic domain, they often play a special role when specialized to (semi-) algebraic varieties. For example, even though fibers are defined for general models, they exhibit a regular behavior in the algebraic context, as mentioned in Section 3.3. Similarly, Morse Theory is especially well-developed over algebraic varieties – especially for distance functions – as mentioned in Section 3.4. This leads to notions such as Euclidean distance degree and data discriminants, which are purely algebraic, and make the theoretical study of the loss landscape more explicit.
>
> From a broad perspective, we see algebraic geometry as one component (albeit an important one) in the toolbox of deep learning theory. We believe that in order to develop a holistic theory, the interactions of many mathematical disciplines is necessary. We acknowledge that the paper would benefit from a discussion around what ideas hold in a more general context, and around the consequences of specializing them to the algebraic domain. We will incorporate the above points in the revised version of the manuscript.
>
> [1] Fefferman, Reconstructing a neural net from its output, 1994.
>
> [2] Vlacic and Bölcskei, Neural network identifiability for a family of sigmoidal nonlinearities, 2019.
>
> [3] Frankle et al., The lottery ticket hypothesis: Finding sparse, trainable neural networks, 2018.

---

### Official Review · Reviewer_XbqL · 2025-03-11

**Significance:** 4
**Argument Clarity:** 4
**Rating:** 5
**Confidence:** 4

**Questions:**

1. The one pitfall of focusing on degree is that it seems to be an unstable invariant in data more generally. In classic regression, there seems to be little empirical difference between fitting data to a degree 5 or a degree 9 polynomial, for example. But this would lead to rather large differences in the covering number estimation. I sometimes think of this as the lack of "stability" of algebraic invariants. Could you address this?

2. Another potential question or concern with the viability of NeuroAG is its utility in studying already trained large NNs, such as LLMs. Of course, when you can design your network architectures to have (piecewise) polynomial activation functions, then you can say alot about them, but if we're focused on discovery of structure in LLMs, it seems like NeuroAG might fall short.

3. Is there any reason Topological Deep Learning (TDL) wasn't mentioned in Section 1.4? See Papamarkou et al from 2024.

4. To strengthen the argument for degree, is there any multiplicative formulation of degrees for composition of layers in a NN? A connection to Bezout's theorem or something similar would strengthen the paper, IMO.

**Discussion Potential:**

4

**Paper Summary:**

This paper argues for the adoption of a new set of tools, stemming from algebraic geometry (AG), to be utilized for the study and design of neural network architectures, or parameterized machine learning more generally. The new subarea of applied algebraic geometry that the authors propose is to be called "neuroalgebraic geometry", which I will abbreviate as NeuroAG.

The authors clarify that the parametric models $f:W \times X \to Y$ to be studied primarily by NeuroAG are those where $W, X$, and $Y$ are all Euclidean spaces and for those mappings $f$ that are polynomials---these are called "algebraic models"---although piecewise polynomial (activation) functions, and even tropical polynomials could be considered as well; the connection to tropical AG is at the end of paper in Section 4. The authors argue that parametric models that lie outside these classes can be approximated by algebraic models; this is discussed in both the introduction and also in Section 4.

The authors note that since machine learning is typically interested in optimizing a parametric model, there are necessarily metrics at play, thus further clarifying that metric algebraic geometry (MAG) is the most pertinent tool set for NeuroAG.

After establishing the scope of their arguments in Sections 1 and 2, the authors proceed to discuss the main tools from AG and MAG that will be useful for NeuroAG in Section 3.

These are

1. Dimension, degree, and covering number---these are basic concepts that can be used to study approximate expressivity and sample complexity for algebraic models.
2. Singularity theory---these are "pinch points" in the neuromanifold that allegedly drive dynamics of the learning process and can introduce implicit bias into training.
3. Fiber theory for the parametrization map of the neuromanifold $M$---this is the image of the parameter space in the function space from $X$ to $Y$ gotten by currying $f:W\times X \to Y$ to $\varphi: W \to M \subseteq Func(X,Y)$. The authors observe that AG provides tools for studying the fibers of $\varphi$, which also connect to invariance properties, such as invariance under the symmetric group. Moreover, studying the set of parameters $w\in \varphi^{-1}(f_w)$ is related to the model identifiability problem.
4. Critical points and Gradient descent: This discusses how studying fiber theory addresses the problem of spurious critical points, which can affect training dynamics. Morse theory is also mentioned when optimizing over the neuromanifold directly (and not using the parametrization $\varphi$), which is more in line with general dynamical systems theory, although the study of data discriminants provides added insight from AG not present in dynamical systems theory.

The paper concludes with an argument that many problems in ML mentioned earlier are connected with deep questions in (tropical and non-tropical) algebraic geometry, thus inviting deeper connections with pure mathematics.

## Update After Rebuttal

It's a strong paper and I recommend acceptance!

**Position:**

Yes

**Position In Title:**

Yes

**Related Work:**

4

**Strengths And Weaknesses:**

I will interpret the position as saying that there are tools that are unique to algebraic geometry that are useful for the study of neural networks and offer insights that can only be accessed via algebraic geometry and not via other areas of pure mathematics, such as geometry, topology or category theory.

The broad support for this argument comes from the observation that many parametric models, including MLPs, CNNs, and NNs with ReLU activation functions, yield neuromanifolds that are actually semi-algebraic sets. Generically these semi-algebraic sets have both a well-defined dimension and degree. Following recent work by Kileel, et al in 2019, this leads to strong bounds (allegedly asymptotic equality) on the covering number of the neuromanifold. This is connected to deep concepts from Cucker and Smale on sample complexity, and, by leveraging the Weyl tube lemma, the approximate expressivity of neural nets. The degree is a concept that is really unique to algebraic geometry, and is not defined in the general continuous/smooth world of mathematics.

Additionally, singularity theory is something which can be difficult in general, but in the algebraic setting there is much tighter control and understanding of singularities. So I find Sections 3.1 and 3.2 particularly convincing.

Finally, I think connections to tropical geometry (Section 4.2) especially intriguing, and again something that is unique to AG.

Of all the cited literature and examples, perhaps the most compelling example for why AG is relevant is the recent result (mentioned on page 4, left column, lines 188-195) that claims that the neuromanifold for CNNs is birationally equivalent to the Segre-Veronese variety. This is really amazing, as are the conjectured connections to transformers.


The weakness of the paper is perhaps in clarifying jurisdictional bounds between algebraic geometry and other flavors of geometry and topology. Perhaps the greatest weakness is in Section 3.3 in the discussion of the fiber-dimension theorem. This is a very general result and one that properly belongs to differential topology. See Guillemin and Pollack (1974 edition, page 21 in the submersions section) on

The Pre-Image Theorem:

If y is a regular value of $f: X -> Y$, then the preimage
$f^{-1}(y)$ is a submanifold of $X$, with $dim f^{-1}(y) = dim X - dim Y$

This result can also be generalized to stratified spaces and mappings, where a map decomposes over piecs of the image as a submersion, so the authors discussion of fibers belongs more properly to the world of differential topology, rather than AG.

This jurisdictional issue persists in Section 3.4 as dynamics is more driven by topology and geometry, over algebra. The data discriminant is the strongest defense against this criticism, but otherwise the sections is really better "owned" by Lefschetz index theorems, Morse theory, Morse homology, Conley index, and other areas of geometry and topology.

**Support:**

4

---

> ### Author Rebuttal · Authors · 2025-03-31
>
> We thank the reviewer for the careful comments and the appreciation.
>
> We wish to comment on the jurisdictional issue raised in the review. We agree that some of the ideas and results we mention extend beyond the realm of algebraic geometry, and belong more naturally to other general theories (e.g., differential geometry or topology). In our view, the multidisciplinary nature of these tools is an advantage for the theory. In fact, we do not wish to argue that algebraic geometry is a universal tool to address the mysteries of deep learning, but rather advocate for adding it to the toolbox of deep learning theorists. We believe that the interaction between different fields of mathematics is necessary to build a solid and holistic theory of deep learning, of which algebraic geometry is one important component. In any case, we agree that it is important to clarify when a result holds more generally, or can be formulated in an alternative language. We will address this in the updated version of the manuscript. We also note that it is often the case that general results yield stronger results in the specific context of algebraic geometry. For example, the generic fibers of an algebraic map not only have the same dimension, but share common properties (at least over the complex numbers), which is important when, e.g., relating fibers to invariances of the model, as mentioned in Section 3.3. Similarly, as mentioned in the review, Morse theory applied in the algebraic context for distance functions leads to tools such as the Euclidean distance degree and data discriminants.
>
> Below, we wish to answer the four questions raised in the review.
>
> 1. We agree that the upper bound on the covering number via the degree (together with the dimension), can be sometimes uninformative in practice, as the “real twistedness” of a variety can be lower than predicted by its degree.  As an immediate future direction to assess this, it might be possible to compute the covering number of the neuromanifold of polynomial CNNs, since they are well understood [2]. Yet, we remark that the degree bounds are general, and are, at the moment, the best ones available. It is possible that in order to obtain better bounds it would be necessary to consider invariants other than dimension and degree, in the spirit of Vitushkin variations [4]. These might lead to estimates that are more “stable” and useful in practice. We will discuss this in Section 3.1 in the updated version of the manuscript.
>
>
> 2. One way we envision to apply neuroalgebraic geometry to a pre-trained model (e.g., an LLM) is via distillation. Once trained, the model can be approximated by an algebraic one (potentially, much smaller), to which algebro-geometric methods apply. This is similar in spirit to mechanistic interpretability [3] – a popular field aiming to interpret pre-trained models (especially LLMs) by distilling them into simple “circuits”. We believe that this is an interesting line for future investigation that might lead to connections between neuroalgebraic geometry and the analysis of models a posteriori – such as interpretability – and we will add a discussion on this in Section 4 in the updated version of the manuscript.
>
>
> 3. We acknowledge that Topological Deep Learning (TDL) is an important emerging field bringing ideas from algebraic topology to machine learning, on a line similar to Geometric Deep Learning. As such, we believe that it deserves mentioning in Section 1.4. We thank the reviewer for suggesting this, and will add a short discussion on TDL in the updated version of the manuscript.
>
>
> 4. At the moment, there is no general formula for the degree of a neuromanifold via the composition of algebraic models (e.g., layers in a NN). Generally speaking, the degree of a (semi-) algebraic variety defined as the image of a polynomial map $\varphi$ (such as neuromanifolds) is difficult to compute, even when $\varphi$ is explicit. Indeed, apart from deep linear networks (see appendix, Section A, especially Eq. 10), the only other algebraic model for which the degree has been computed are polynomial CNNs, relying on the known formula for the degree of the Segre-Veronese. In this case, the degree is indeed multiplicative (see [2], Eq. 11), in the sense that adding a layer increases it by a factor depending on the filter size and degree of the activation function. In particular, there is no degree formula for deep polynomial MLPs. Finding a general formula is, in our opinion, an important research question, which would provide insights into the effects of depth (in a deep network) on sample complexity.
>
>
> [1] Zhang and Kileel, Covering Number of Real Algebraic Varieties and Beyond: Improved Bounds and Applications, 2024.
>
> [2] Shahverdi et al., On the Geometry and Optimization of Polynomial Convolutional Networks, 2024.
>
> [3] Bereska et al., Mechanistic Interpretability for AI Safety--A Review, 2024.
>
> [4] Friedland and Yomdin, Vitushkin-Type Theorems, 2013.

---

> > ### Comment · Reviewer_XbqL · 2025-04-05
> >
> > thank you for the comment, i look forward to reading the revision.

---

### Official Review · Reviewer_NoCV · 2025-03-14

**Significance:** 2
**Argument Clarity:** 3
**Rating:** 4
**Confidence:** 4

**Questions:**

### Suggestions/questions:

* As someone without any expertise in the field of algebraic geometry it would be very useful to have a roadmap, case studies, or some basic examples of how computing singularities/fibers of the parameterisation led to tangible outcomes.

* Some remarks on the connection of this field to that of topological data analysis, or neural homology would be useful, or clarifications of the distinction of this approach with these established paths. Some references: https://arxiv.org/abs/2502.01360 , https://arxiv.org/abs/1802.04443 , https://run.unl.pt/bitstream/10362/129615/1/TAA0115.pdf

**Discussion Potential:**

3

**Paper Summary:**

The paper advocates for the study of machine learning models, particularly neural networks, through the lens of algebraic geometry. It introduces "neuroalgebraic geometry" as a framework to analyze algebraic models (e.g., networks with polynomial activations), whose function spaces form semi-algebraic varieties. The authors outline a dictionary linking algebro-geometric invariants (dimension, degree, singularities) to machine learning concepts (sample complexity, expressivity, training dynamics). Contributions include theoretical connections, a review of related approaches (e.g., information geometry, kernel methods), and discussions on extending results to non-algebraic models via approximation or tropical geometry. The position emphasizes the potential of algebraic geometry to advance deep learning theory and inspire interdisciplinary research.

---

# Update after rebuttal

I have revised my score to Accept after seeing author's comments.

**Position:**

Yes

**Position In Title:**

No

**Related Work:**

3

**Strengths And Weaknesses:**

### Strengths:

* Clear articulation of a novel interdisciplinary position, bridging algebraic geometry and machine learning.
* Comprehensive literature review, integrating diverse fields (algebraic geometry, singular learning theory, tropical geometry).
* Well-structured argument with a detailed "dictionary" (Table 1) connecting geometric invariants to ML concepts.
* Addresses alternative views (e.g., critiques of algebraic models’ practicality) and related approaches (information geometry, neural tangent kernels).

### Weaknesses

While the paper is clearly well-written, and while it argues a salient, well-referenced points, it is not directly clear from paper's contents what immediate, practical contribution/insight can follow from these theoretical advances. This is not to say that none exist, but rather - that these need to be much better argued, especially in the context of modern LLM development and large-scale models of today. Or, contrary - perhaps authors wish to argue that this will enable smaller models to exist? It is not entirely clear.

There are technical components of this approach to mention, such as heavy reliance on polynomial activations (limiting applicability), but that is something authors acknowledge, but perhaps the main point here is that a more clear highlight of open problems, or actionable research directions for the ML community would be very useful.

In addition to interaction with LLMs, one might ask how this mathematical framework interacts with higher arity data (such as that present in molecular analysis, or social network analysis), or data which is defined recursively (such as abstract syntax trees or structured data often present in code). It is not immediately clear whether algebraic geometry (as opposed to other, similar branches of mathematics) is best suited for studying manifolds usually occuring with these data modalities.

**Support:**

3

---

> ### Author Rebuttal · Authors · 2025-03-31
>
> We thank the reviewer for the constructive feedback.
>
> We wish to comment on the practical consequences of the theoretical tools discussed in our work. Generally speaking, our perspective is that developing a fundamental theory of machine learning – which is the goal of neuroalgebraic geometry and of similar approaches – can eventually lead to an informed design of models and methods satisfying desirable properties, resulting in practical benefits. For example:
> - Understanding the precise connection between dimension, degree and sample complexity (Section 3.1 and in the appendix) would provide criteria for designing neural architectures with controllable scalability properties.
> - Understanding the singularities of neuromanifolds and their connection to subnetworks (Section 3.2) would provide criteria for designing neural architectures which are implicitly biased towards simpler models (i.e., smaller networks), potentially resulting in benefits in terms of robustness and generalization.
> - Understanding the dynamical invariants of gradient descent (Section 3.4) would provide insights into initialization, with benefits in terms of stability of the training process.
>
> As a research plan for neuroalgebraic geometry, we envision studying the modules and components commonly deployed in deep learning first one by one, and then to investigate their interactions. Recently, the neuroalgebraic geometry of modules such as convolutions [1] and self-attention mechanisms [2] has been understood, bridging the gap between theory and contemporary models (e.g., the attention-based transformers in LLMs). Yet, several components and their interactions are still unexplored. Examples of actionable research directions are:
> - **Skip connections**. Since an algebraic model with skip connections remains algebraic, neuroalgebraic geometry is suitable in this context. Skip connections might be related to the smoothing of the neuromanifold, as suggested by [3].
> - **Graph Neural Networks** (GNNs) are popular across several domains and closely related to both CNNs and self-attention mechanisms. Therefore, the results from [1, 2] might be extendable to GNNs.
> - **State Space Models** (SSMs) are efficient alternatives to transformers for natural language modelling [4] and their layers are defined in a purely algebraic way. Thus, neuroalgebraic geometry is applicable and might provide insights into how these models compare to transformers.
>
> Next, we wish to discuss the point raised in the review regarding structured data, which we find interesting. Additional structure in the data –  such as graph structures and recursion in sequences – is typically incorporated in the design of the neural architecture, which is in turn reflected by the geometry of the neuromanifold. This principle lies at the heart of Geometric Deep Learning, which focuses on neural architectures tailored on geometric structures of data. For example, convolutional networks incorporate data symmetries (e.g., in images), resulting in neuromanifolds with simple parametrizations and mild singularities [1]. Similar considerations hold for self-attention mechanisms [2]. GNNs incorporate analogous permutation symmetries of graph-structured data, while recurrent architectures such as SSMs incorporate recursive structures of sequences. Studying the neuromanifolds of these existing models might lead to overall insights on how to design architectures that exploit structured data.
>
> We believe that the above considerations on practical insights, research directions and structured data are important, and we will incorporate them in the updated version of the manuscript.
>
> Lastly, we comment on the relation to Topological Data Analysis (TDA). The latter studies (the topology of) the data manifold and, as showcased by the works mentioned in the review, has been applied to analyze the latent data representations induced by deep networks, i.e., the image of the data manifold in the hidden layers of the networks. In contrast, neuroalgebraic geometry is concerned with (the geometry of) the function space parametrized by machine learning models. Therefore, the main difference between the two approaches is that they focus on different spaces, both of which are relevant in deep learning. From a broader perspective, however, both approaches aim at bridging geometry and topology – specifically, tools of algebraic flavour – with machine learning and data analysis. Therefore, we believe that TDA is important to mention among the related approaches in Section 1.4, and we will address this in the revised version of the manuscript.
>
>
>
> [1] Shahverdi et al., On the geometry and optimization of polynomial convolutional networks, 2024.
>
> [2] Henry et al., Geometry of lightning self-attention: Identifiability and dimension, 2024.
>
> [3] Orhan and Pitkow, Skip connections eliminate singularities, 2017.
>
> [4] Gu eta al., Mamba: Linear-time sequence modeling with selective state spaces, 2023.

---

> > ### Comment · Reviewer_NoCV · 2025-04-06
> >
> > I thank the authors for their explanations and comments. I have revised my score to Accept

---

### Decision · Program_Chairs · 2025-04-30

**Decision:**

Accept (spotlight poster)

**Comment:**

The paper was reviewed by four expert reviewers.

The major strengths of the paper are on its novel interdisciplinary position and well-structured and convincing argument. The paper introduces ``neuroalgebraic geometry'' for bridging algebraic geometry and machine learning and outlines a dictionary linking algebro-geometric invariants to machine learning concepts. It also provides a detailed ``dictionary,'' connecting geometric invariants to machine learning concepts. The paper presents a comprehensive literature review, integrating diverse fields, such as algebraic geometry, singular learning theory, and tropical geometry. The paper addresses alternative views and discusses critiques of algebraic models' practicality. As a result, the paper has a high potential to inspire discussions in the community.

On the other hand, there are some weaknesses pointed out by the reviewers. The paper heavily relies on polynomial activations, and it limits its applicability. Also, it makes the paper more comprehensive if the jurisdictional bounds between algebraic geomtery and other branches of mathematics is included. In addition, the paper does not clearly state polynomial optimization and related hierarchies of convex programs.

The authors' rebuttal was read and considered by the reviewers. Over the author-reviewer discussion, most of the concerns were clearly addressed. As a result, the reviewers were all positive about the paper. The AC agreed with the reviewers' opinions and reached this decision.